# Tube Loss: A Novel Approach for High Quality Prediction Interval Estimation

## Abstract

This paper proposes a continuous loss function termed 'tube loss' for Prediction Interval (PI) estimation. The minimizer of the proposed tube loss is a pair of functions $\mu_1(x)$ and $\mu_2(x)$ such that the interval $[\mu_2(x), \mu_1(x)]$ contains $t$ fraction of $y_i$ values. The tube loss function also facilitates an upward or downward movement of the PI tube so that the estimated PI may cover the densest regions of response values, thus allowing the sharpening of the width of PI, especially when the distribution of the response is skewed. The tube loss function-based machine learning models also have the privilege of trading off the calibration error and the width of PI by solving a single optimization problem. We have illustrated the use of tube loss functions in kernel machines, neural networks, and sequential deep learning models. Our numerical experiments show that the tube loss function is effective in yielding narrow and more accurate PIs compared to the existing methods.

## 1 Introduction

Given the training set $T = \{(x_i, y_i) : x_i \in \mathbb{R}^n, y_i \in \mathbb{R}, i = 1, 2, ..., m\}$, a standard regression model targets the estimation of $E(y/x)$. But often times, the interest is to predict the value of $y$, the dependent variable, given a value of the independent variable $x$. In such cases, an interval of values of $y$ with certain confidence is offered as an estimate of $y$ for a given $x$. Such intervals are called the Prediction Intervals (PI) of $y$ given $x$.

For a given $x$ and confidence $t \in (0, 1)$, a PI of $y$ is an interval $[\mu_2(x), \mu_1(x)]$ such that $P(\mu_2(x) \leq y \leq \mu_1(x)|x) \geq t$. Ideally, a PI should have the following desirable properties. First, the confidence associated with the PI should be greater than equal to $t$, and second, among the PIs with confidence greater than equal to $t$, the one with the minimum average width is the optimal one. Clearly, the confidence of a PI is an increasing function of its width. Thus one needs to trade-off between the two by minimizing the average width subject to the condition that its confidence is greater than equal to $t$.

Several approaches are available in the literature for finding PI. Some of these like the delta method (Hwang & Ding (1997)Chryssolouris et al. (1996)), the mean-variance estimation method (Nix & Weigend (1994)) and the Bayesian technique (MacKay (1992)) assume the noise to be normally distributed and homogeneous. Such models may perform poorly in many real-life applications when these assumptions do not hold. Some of the methods like (Chung et al. (2021)) and (Cui et al. (2020)) first estimate the $E(y/x)$ and then attempt to approximate the error distribution function or its different moments for the PI estimation.

A popular method for finding PI is through the estimation of quantile functions. For finding a PI with $t$ confidence, one simply requires independent estimates of $q^{th}$ and $(q + t)^{th}$ quantile functions, say, $F_q(x)$ and $F_{q+t}(x)$, respectively. Evidently, the PI is given by $[F_q(x), F_{q+t}(x)]$. In non-parameteric framework, the estimation of $q^{th}$ quantile function $F_q(x)$ is obtained by minimizing the pinball loss function given by

$$\rho_\tau(u) = \begin{cases} u, & \text{if } u \geq 0, \\ (\tau - 1)u, & \text{otherwise,} \end{cases} \tag{1}$$

with $\tau = q$. But, the pinball loss function can only ensure *average calibration* and fails to obtain the *individual calibration* (Chung et al. (2021)).

Also, in a non-parametric framework, the estimation of PI through quantiles start from arbitrarily fixing a $q$ value such that $q + t \leq 1$ and then estimating the functions $F_q(x)$ and $F_{q+t}(x)$ by solving two different optimization problems independently. Therefore, it cannot minimize the width of the resulting PI and naturally fails to ensure that the PI tube should contain more dense regions of target values.

At first, Khossravi et al. propose a loss function for direct estimation of PI in neural network framework. They develop Lower Upper Bound Estimation (LUBE) Khosravi et al. (2010) method which simultaneously estimates the lower and upper functions $\mu_2(x)$ and $\mu_1(x)$, and thus allow minimization of the PI width. The loss function used in the LUBE method is the *zero-one loss function*, which is equal to one if the response $y_i$ lies in the interval $[\mu_2(x), \mu_1(x)]$, otherwise equal to zero. The LUBE method minimizes the *zero-one loss function* in its optimization problem in such a way that $t$ fraction of $y_i$ lies in the interval $[\mu_2(x), \mu_1(x)]$. But, the *zero-one loss function* is a non-continuous function which makes the LUBE optimization problem difficult to be solved efficiently. The LUBE method uses the Simulated Annealing (SA) (Kirkpatrick et al. (1983)) to solve its optimization problem but, the solution obtained may be sub-optimal and lack precision.

Pearce et al. have improved the LUBE method in their work (Pearce et al. (2018)) by using the Quality-Driven distribution free (QD) loss function. They have approximated the *zero-one loss function* with a sigmoid function and solved the optimization problem with the gradeint descent method. They were able to obtain smoother and better PI than the LUBE method in terms of reliability and length of the interval. But, the efficacy of their method depends upon the quality of the approximation of *zero-one loss function*.

This paper proposes a loss function for PI estimation which we call 'tube loss'. The proposed loss function has several novel features. First, it is a continuous function unlike the others used for PI estimation previously, thus allowing efficient application of gradient descent method for estimation of PI without needing an approximation of the loss function by a continuous function. Second, the proposed loss function allows simultaneous estimation of a pair of functions $\mu_2(x)$ and $\mu_1(x)$ defining the PI. For a given training set, $T = \{(x_i, y_i) : x_i \in \mathbb{R}^n, y_i \in \mathbb{R}, i = 1, 2, ..., m\}$, we show that the minimizer ($\mu_2(x)$, $\mu_1(x)$) of the average tube loss ensures confidence $t$ for the estimated PI as m becomes large. Finally, the proposed loss function has a tuning parameter $r$. By changing $r$, the PI tube can be moved up or down help capturing denser regions of the response values leading to reduced width of the interval. Also, we introduce an additional tuning parameter $\delta$ for balancing the calibration and the overall width of PI while solving the optimization problem. This is used for re-calibrating the model for lowering PI width, when the observed coverage is larger than target $t$.

We implement the proposed tube loss function in kernel machines (Vapnik (1999)), Neural Networks and deep learning frameworks. In the kernel machines, the tube loss-based machine is always preferable than the existing quantile loss-based machine. Unlike the quantile-based Support Vector Machine (SVM), which requires estimation of $\mu_2(x)$ and $\mu_1(x)$ separately, the PI estimation based on the proposed loss function allows trade off between the width of the PI and its coverage while solving the optimization problem. Since the quantile-based machine requires solving two optimization problems for estimation of the $q^{th}$ and $(t + q)^{th}$ quantiles compared to solving one optimization problem for the tube loss-based PI estimation, it requires more time for training. As a consequence the searching for a narrow PI with a given confidence $t$ is easier for the latter.

In the NN framework, we compare the tube loss-based NN model with the best-performing QD loss-based model (Pearce et al. (2018)). Unlike their model, the tube loss-based NN model minimizes a continuous loss function and thus, does not require any approximation. Consequently, it makes the tube loss-based NN estimation more accurate than the QD loss-based model (Pearce et al. (2018)). Also, as stated above, by adjusting the tuning parameter $r$, the tube loss function enables the movement of the PI tube for capturing the denser regions of $y_i$, and thus, reducing the width, which is not the case with QD loss function.

We have also designed the LSTM model (Hochreiter & Schmidhuber (1997)) with the tube loss function and use it for obtaining an efficient probabilistic forecast for time-series data set. We show that the tube loss-based LSTM model gives reliable PI.

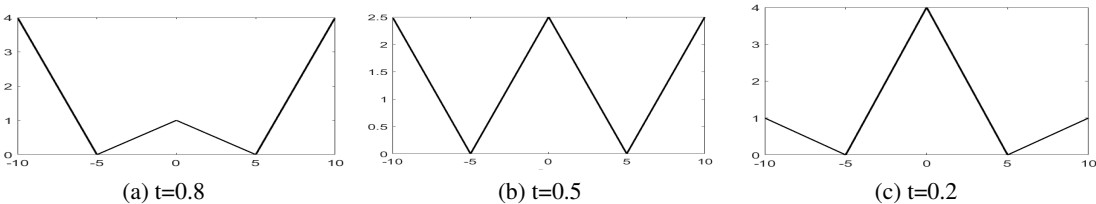

Figure 1: Tube Loss with $\mu_2 = -5$ and $\mu_1 = 5$.

## 2  TUBE LOSS FUNCTION

For given $t \in (0, 1)$, we propose the following loss function for estimating PI with confidence $t$, say, $[\mu_2, \mu_1]$, where $\mu_2 \leq \mu_1$ ,

$$
L_t(y, \mu_1, \mu_2) = \begin{cases}
t(y - \mu_1), & \text{if } \mu_1 < y, \\
(1 - t)(\mu_1 - y), & \text{if } \mu_2 \leq y_i \leq \mu_1 \text{ and } y_i \geq r\mu_1 + (1 - r)\mu_2, \\
(1 - t)(y - \mu_2), & \text{if } \mu_2 \leq y_i \leq \mu_1 \text{ and } y_i < r\mu_1 + (1 - r)\mu_2, \\
t(\mu_2 - y), & \text{if } \mu_2 > y,
\end{cases}
\tag{2}
$$

where $r > 0$ is a user-defined parameter.

Let $Y = \{y_1, y_2, ...., y_m\} \subset \mathbb{R}$ and $t \in (0, 1)$. Let us define the following subsets of $\mathbb{R}$: $\Re_1(\mu_1, \mu_2) = \{y : y > \mu_1\}$, $\Re_2(\mu_1, \mu_2) = \{y : \mu_2 < y < \mu_1, y > r\mu_1 + (1 - r)\mu_2\}$, $\Re_3(\mu_1, \mu_2) = \{y : \mu_2 < y < \mu_1, y < r\mu_1 + (1 - r)\mu_2\}$ and $\Re_4(\mu_1, \mu_2) = \{y : y < \mu_2\}$. Notice that the sets $\Re_i(\mu_1, \mu_2), i = 1, 2, 3, 4$ are so defined that they do not include the points on the boundaries.

**Preposition 1.** The minimizer $(\mu_{1t}, \mu_{2t})$ of $\sum_{i=1}^m L_t(y_i, \mu_1, \mu_2)$ with respect to $(\mu_1, \mu_2)$ satisfies
(i) $\frac{m_1 + m_4}{m_2 + m_3} \to \frac{1-t}{t}$,
(ii) $\frac{m_1}{m_2} \to \frac{1-t}{t}$,
(ii) $\frac{m_4}{m_3} \to \frac{1-t}{t}$,
as $m \to \infty$, where $m_k$ denotes the number of $y_i$'s in $\Re_k(\mu_{1t}, \mu_{2t})$, $k = 1, 2, 3, 4..$

Proof: Consider $(\mu_1, \mu_2) = (\mu_{1t} + \delta_1, \mu_{2t} + \delta_2)$, say, $(\mu_1(\delta_1), \mu_2(\delta_2))$, where $\delta_1$ and $\delta_2$ $((\delta_1, \delta_2) \neq (0, 0))$ are so chosen that $\{y : y \in \Re_k(\mu_{10}, \mu_{20})\}$ is equal to $\{y : y \in \Re_k(\mu_1(\delta_1), \mu_2(\delta_2))\}$. Clearly we have $\sum_k m_k \leq m$. Now we compute the change in the value of the loss function $D(\delta_1, \delta_2) = \sum_{i=1}^m L_t(y_i; \mu_{10}, \mu_{20}) - \sum_{i=1}^m L_t(y_i; \mu_1(\delta_1), \mu_2(\delta_2))$ which is always non-negative for the following choices of $(\delta_1, \delta_2)$. The following results follow assuming that $y_i$'s are the realizations of a continuous random variable with no discrete probability mass.

(i) Notice that for $\delta > 0$,

$$
D(\delta, -\delta) = -m_1 t + m_2(1 - t) + m_3(1 - t) - m_4 t \geq 0,
\tag{3}
$$

and

$$
D(-\delta, \delta) = m_1 t - m_2(1 - t) - m_3(1 - t) + m_4 t \geq 0.
\tag{4}
$$

As $m \to \infty$ (which in turn imply $m_i \to \infty, i = 1, 2, 3, 4$) the equations (3) and (4) imply that,

$$
\frac{m_1 + m_4}{m_2 + m_3} \to \frac{1 - t}{t}.
\tag{5}
$$

(ii) Similarly, as $m \to \infty$, $D(\delta, 0) \geq 0$ and $D(-\delta, 0) \geq 0$ imply that

$$\frac{m_1}{m_2} \to \frac{1-t}{t} \tag{6}$$

and $D(0, \delta) \geq 0$ and $D(0, -\delta) \geq 0$ imply that

$$\frac{m_4}{m_3} \to \frac{1-t}{t}. \tag{7}$$

Suppose for some $t \in (0, 1)$, given a training set $T = \{(x_1, y_1), ...., (x_l, y_l)\}$ where $x_i \in \mathbb{R}^n$ and $y_i \in \mathbb{R}$, the problem is to minimize the empirical risk $\frac{1}{m} \sum_{i=1}^{m} L_t(y_i, \mu_1(x_i), \mu_2(x_i))$ with respect to $(\mu_1(x), \mu_2(x))$. Let $m_k$ denote the cardinality of the set $\Re_k(\mu_{1t}(x), \mu_{2t}(x))$, $(k = 1, 2, 3, 4)$ where $(\mu_{1t}(x), \mu_{2t}(x))$ is the minimizer of the empirical risk defined above. We then have the following proposition.

**Preposition 2.** For $t \in (0, 1)$ as $m \to \infty$,

(i) $\frac{m_1 + m_4}{m_2 + m_3} \to \frac{1-t}{t}$,
(ii) $\frac{m_1}{m_2} \to \frac{1-t}{t}$,
(ii) $\frac{m_4}{m_3} \to \frac{1-t}{t}$,

with probability 1, provided $(x_i, y_i), i = 1, 2, ..., m$ are iid following a distribution $p(x, y)$ with $p(y|x)$ is absolutely continuous almost everywhere.

Proof: The proof follows from (Schölkopf et al. (2000)).

**Note 1.** With decrease in the the value of the tuning parameter $r$, the separating curve $(r\mu_1(x) + (1 - r)\mu_2(x))$ of the PI tube moves downward inside the PI tube causing $\frac{m2}{m3}$ to increase but ensuring that $m_k$'s satisfy the conditions stated in Proposition 2, thus helping in capturing the denser part of the $y_i$'s keeping the confidence at the prefixed level $t$. On a similar line, the separating curve can be moved up by increasing the value of $r$.

**Note 2.** Since the tube loss function enables simultaneous estimation of $\mu_1(x)$ and $\mu_2(x)$, the width of the estimated tube can be traded-off against the calibration in a single optimization problem as stated below:

$$\min_{(\mu_1, \mu_2)} \sum_{i=1}^{m} L_t(y_i, \mu_1(x), \mu_2(x)) + \delta \sum_{i=1}^{l} |\mu_1(x) - \mu_2(x))|. \tag{8}$$

In practice, however, given a training set T, we obtain $[\bar{\mu}_1(x), \bar{\mu}_2(x)]$ by minimizing only the tube loss function. If the confidence of the estimated PI tube $[\bar{\mu}_1(x), \bar{\mu}_2(x)]$ is found to be more than the target $t$, then we solve the problem 8 for a suitably chosen $\delta(> 0)$ for tuning it further to reduce the average width of the estimated PI achieving confidence $t$.

## 3 PI ESTIMATION USING THE TUBE LOSS

In this section, we discuss the use of tube loss function for PI estimation in different learning frameworks.

### 3.1 TUBE LOSS BASED PI ESTIMATION IN KERNEL MACHINES

Unlike quantile-based kernel machines, the tube loss-based kernel machine requires the solution of a single optimization problem. It obtains the two functions $\mu_1(x)$ and $\mu_2(x)$

$$\mu_1(x) := K(A^T, x)\alpha + b_1 \text{ and } \mu_2(x) := K(A^T, x)\beta + b_2 \tag{9}$$

simultaneously. We also term the tube loss-based kernel machine as the 'Support Vector Prediction Interval'(SVPI) model, which seeks the solution to the problem

$$\min_{(\alpha,\beta,b_1,b_2)} J(\alpha,\beta,b_1,b_2) = \frac{\lambda}{2l}(\alpha^T\alpha + \beta^T\beta) + \frac{1}{l}\sum_{i=1}^{l} L_t\big(y_i, \big(K(A^T,x_i)\alpha + b_1\big), \big(K(A^T,x_i)\beta + b_2\big)\big)$$
$$+ \delta\sum_{i=1}^{l}\big|(K(A^T,x_i)(\alpha - \beta) + (b_1 - b_2)\big|, \tag{10}$$

where the regularization parameter $\lambda$, the tube shift parameter $r$ and the tube width parameter $\delta$ are all positive and user defined. We solve the problem (10) using the gradient descent method.

The tuning parameter $r$, as stated above, enables capturing the denser region of response values by moving the separator curve up or down. An idea about the skewness of the distribution $y_i/x$ may act as a guide to choosing the value of the tuning parameter $r$.

### 3.2 Tube loss based PI estimation in Neural Network

The tube loss-based NN model requires the solution of the problem (8), where for given $x \in \mathbf{R}^n$, $\mu_1(x)$ and $\mu_2(x)$ are two outputs of the network.

### 3.3 Tube loss in LSTM method for probabilistic forecasting

In the distribution-free setting, the most popular method for obtaining the probabilistic forecast for a given time-series signal is to use the quantile loss. For obtaining PI with confidence $t$, the researchers train a pair of LSTM models for estimating the $\tau$-th and $(t + \tau)$-th quantiles. But, with the tube loss function, for estimating the PI one needs to train only one LSTM model. Thus, the tube loss-based LSTM is less computationally expensive and facilitates sharpening the width of PI more efficiently.

## 4 Experimental Results

In this section, we shall perform experiments to observe the effectiveness of the proposed tube loss function in kernel machines, neural networks and sequential deep learning models. But, before presenting the numerical results, we detail below the popular evaluation criteria that are used for assessing the quality of PI.

(a) Prediction Interval Coverage Probability (PICP): The fraction of $y_i$ which lie inside of the PI tube.

(b) Mean Prediction Interval Width (MPIW): The mean width of estimated PI given by $\frac{1}{m}\sum_{i=1}^{m}(\mu_1(x_i) - \mu_2(x_i))$.

(c) Error: It is the coverage error $|t - PICP|$ where target $t$ is the target coverage of PI.

### 4.1 Tube loss in Kernel machines

We simulate the two artificial datasets by generating the independent variable $x_i$ from $U(0,1)$ and response values $y_i$ using the relation

$$y_i = \frac{sin(x_i)}{(x_i)} + \epsilon_i \tag{11}$$

In dataset **A**, the noise $\epsilon_i$ is added from $\mathcal{N}(0, 0.8)$. In dataset **B**, the noise $\epsilon_i$ is added from asymmetric $\chi^2(3)$. We generated 500 data points for the training set and 1000 data points for the testing set for both datasets.

For target $t = 0.8$ and $t = 0.9$, we train our tube loss-based SVPI model with linear kernel with parameter values $\lambda = 1$, $r = 0.5$ and $\delta = 0$ and show the estimated PI on the test set in Figure (2). The estimated PI tubes contain 0.799 and 0.909 fractions of $y_i$ values with 2.10 and 2.78 MPIW respectively.

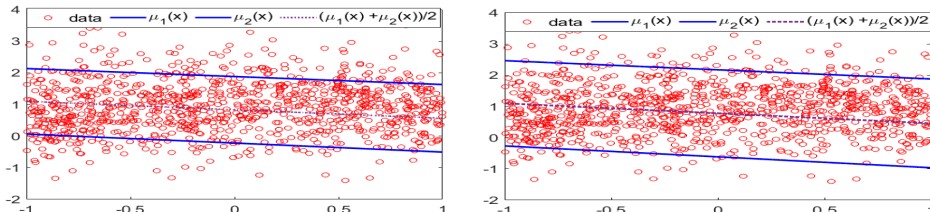

Figure 2: Tube loss based SVPI estimation for (a) $t = 0.8$ and (b) $t = 0.9$

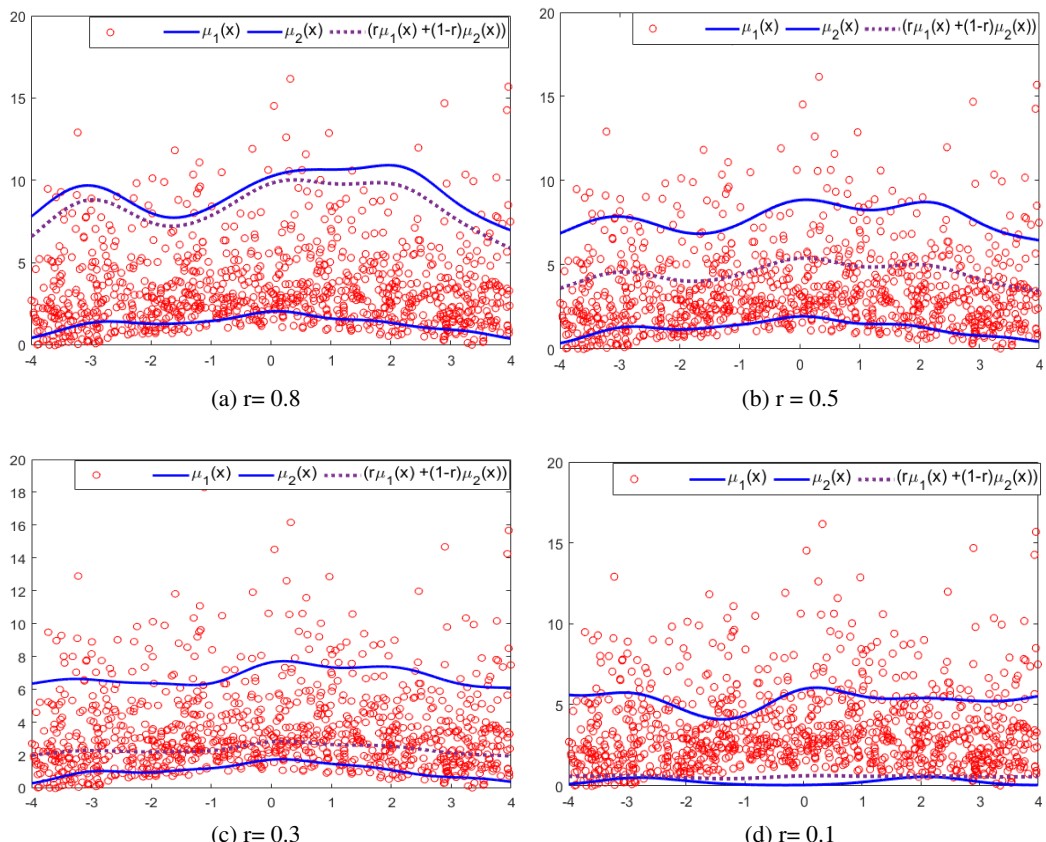

Figure 3: Location of PI tube changes with r values in Tube loss based SVPI model.

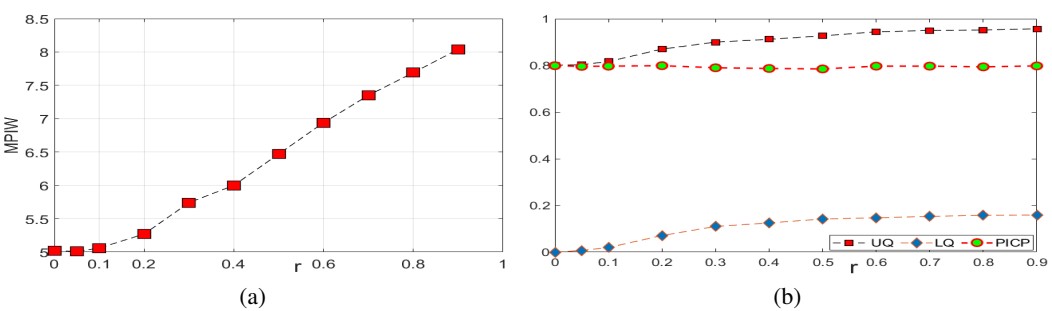

Figure 4: Plot of (a) r against MPIW ,(b) PCIP, UQ and LQ for dataset **B**.

| Dataset | PICP | MPIW | Error | Time(s) |
|---------|------|------|-------|---------|
| A | $0.805 \pm 0.018$ | $2.156 \pm 0.069$ | $0.015 \pm 0.011$ | 61.50 |
| B | $0.601 \pm 0.028$ | $3.174 \pm 0.165$ | $0.0210 \pm 0.018$ | 41.55 |

Table 1: Tube Loss based SVPI model on dataset **A** and **B**

| Dataset | Upper Quantile | Lower Quantile | PICP | MPIW | Error | Time (s) |
|---------|----------------|----------------|------|------|-------|----------|
| A | 0.95 | 0.15 | $0.78 \pm 0.017$ | $2.088 \pm 0.109$ | $0.022 \pm 0.015$ | 219.38 |
| | 0.90 | 0.10 | $0.78 \pm 0.019$ | $2.002 \pm 0.068$ | $0.022 \pm 0.016$ | 221.72 |
| | 0.85 | 0.05 | $0.79 \pm 0.025$ | $2.151 \pm 0.078$ | $0.021 \pm 0.016$ | 217.38 |
| B | 0.90 | 0.30 | $0.59 \pm 0.032$ | $4.6609 \pm 0.153$ | $0.8713 \pm 0.056$ | 138.05 |
| | 0.95 | 0.35 | $0.58 \pm 0.023$ | $5.5369 \pm 0.272$ | $0.5012 \pm 0.049$ | 146.60 |
| | 0.80 | 0.20 | $0.59 \pm 0.027$ | $3.6196 \pm 0.143$ | $1.2461 \pm 0.079$ | 146.41 |

Table 2: Quantile loss based SVQR model on dataset **A** and **B**.

In dataset **B**, we target to estimate the PI for $t = 0.8$ with RBF kernel. In Figure 3 (b), the SVPI model with parameters $r = 0.5$ and $\delta = 0$ obtains the PICP 0.79 and MPIW 6.47. However, the estimated PI tube does not capture the densest region of response values leading to large MPIW values as noise in the data is from the asymmetric $\chi^2$ distribution. For this, we need to shift the estimated PI tube downwards. Figure 3 (a) (c) and (d) shows the PI obtained by the tube loss-based SVPI model for $r = 0.8, 0.3$, and $0.1$. As we decrease the $r$ values, the estimated PI tube moves downward, capturing denser regions of $y$ values with lower MPIW. We have plotted the MPIW, PICP, LQ, and UQ obtained by the SVPI model against different $r$ values in Figure 3. The LQ (Lower Quantile) and UQ (Upper Quantile) are the fractions of $y_i$ lying below the upper and lower functions of our estimated PI tube. The LQ and UQ values decrease with the decrease in r values leading to a downward movement of the PI tube with lesser MPIW values without compromising the PICP of estimate. The MPIW improves significantly with the decrease in $r$ values. In fact, MPIW values improve by $21.80\%$, if we decrease the $r$ from $0.5$ to $0.1$.

We generate the training and testing data sets **A** and **B** in ten different trials. At first, we check the performance of the tube loss-based SVPI model with $\delta = 0$ and quantile loss-based SVQR model and report the numerical results in Table 1 and 2 respectively. The target PI was set to 0.80 and 0.6 for dataset **A** and **B** respectively. In our experiments, the tube loss-based SVPI model outperforms the quantile loss-based SVQR model. The significant advantage of the proposed SVPI model over SVQR model is improvement in training time as the SVQR model needs to be trained twice to obtain $\tau$ and $t + \tau$ quantiles.

Since the tube loss-based SVPI model obtains larger PICP than target $0.80$ on dataset **A**, we recalibrate the SVPI model with $\delta = 0.001, 0.002, 0.005, 0.008$ and $0.01$ for improving the MPIW values. Figure 5 shows that $\delta$ can be used for improving the MPIW by adjusting PICP in SVPI model. At $\delta = 0.002$, we manage to obtain the $0.801$ PICP and MPIW of $2.142$.

## 4.2 TUBE LOSS IN NEURAL NETWORK

Next, we use the proposed tube loss function in the neural network to assess its validity. In the NN framework, the LUBE method is the most popular and established method for PI estimation. But,

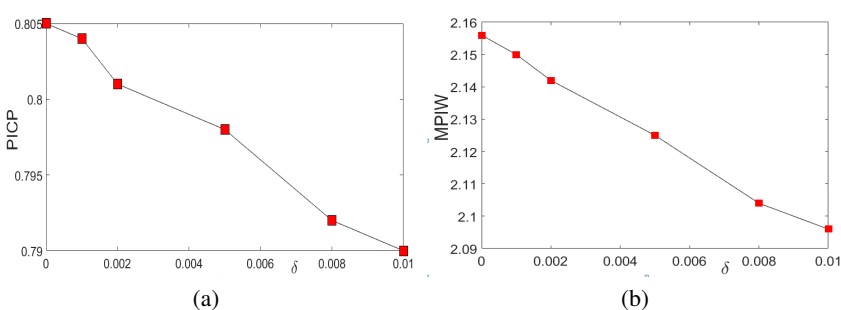

Figure 5: Plot of (a) $\delta$ against PICP ,(b) $\delta$ against MPIW.

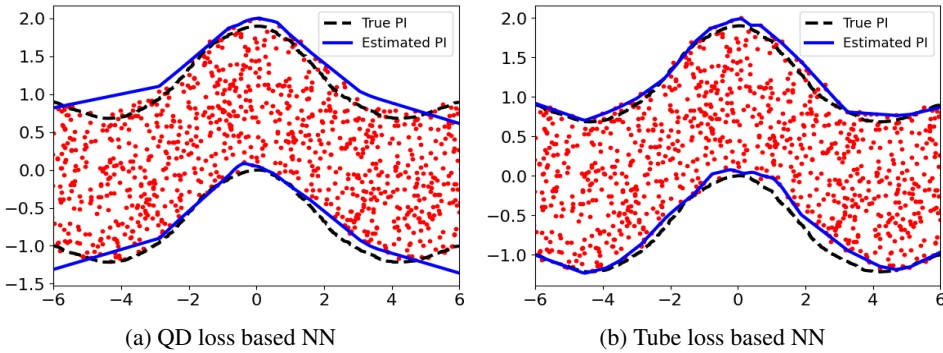

(a) QD loss based NN  (b) Tube loss based NN

Figure 6

in Pearce et al. (2018), authors have shown that their QD loss-based NN is more efficient in terms of smoothness, coverage probability, and width of PI as they are using the gradient-based technique for solving the optimization problem similar to LUBE. Therefore, we shall be comparing our tube loss-based NN model with the recent QD loss-based NN model.

We perform two experiments with simulated artificial datasets that clearly show the tube loss-based NN can obtain a better estimate of PI over the existing zero-one loss-based QD loss function. For these experiments, we have considered 100 neurons in first hidden layer with 'ReLU' activation function and 'Adam' optimizer.

At first, we generate 1000 data points by adding uniform noise to the sine function as shown below

$$y_i = \frac{sin(x_i)}{x_i} + \epsilon_i, \tag{12}$$

where $x_i$ is $U(-2\pi, 2\pi)$ and $\epsilon_i$ is $U(-1, 1)$. We use both the Tube Loss-based NN model and the existing QD loss-based NN model to obtain a PI with confidence 0.95. Figure 6 shows the upper and lower limits of the PIs based on the Tube Loss-based NN model, QD loss-based NN model and the true PI obtained by finding the $97.5\%$ and $2.5\%$ quantiles of the distribution of $y_i$ given $x_i$. Evidently, the tube loss-based NN model approximates the truth PI better than the QD loss-based NN model.

Next, we generate a data set comprising 1000 data points with heteroscedastic noise by sampling $x_i$ from $U(-2, 2)$ and then generating $y_i$ as follows

$$y_i = 0.3sin(\pi x) + \epsilon_i, \tag{13}$$

where $\epsilon_i$ from $N(0, x_i^4)$.

Like Figure 6, Figure 7 shows the limits of the PIs based on tube loss-based NN model and QD loss-based NN model and the true PI with $95\%$ confidence. Clearly the tube loss-based NN model approximates the true PI better than the QD-based NN model.

A probable reason for the worse performance of the QD loss-based NN model is its inability to well approximate the zero-one loss function with the sigmoidal functions.

|  | PICP | | MPIW | |
|---|---|---|---|---|
|  | QD Loss | Tube Loss | QD Loss | Tube Loss |
| Boston | $0.92 \pm 0.01$ | $0.95 \pm 0.02$ | $1.16 \pm 0.02$ | $1.18 \pm 0.07$ |
| Concrete | $0.94 \pm 0.01$ | $0.95 \pm 0.00$ | $1.09 \pm 0.01$ | $1.36 \pm 0.02$ |
| Energy | $0.97 \pm 0.01$ | $0.95 \pm 0.01$ | $0.47 \pm 0.01$ | $0.65 \pm 0.06$ |
| Wine | $0.92 \pm 0.01$ | $0.95 \pm 0.01$ | $2.33 \pm 0.02$ | $2.81 \pm 0.03$ |

Table 3: Performance of the QD loss and proposed Tube based NN models on benchmark datasets.

BENCHMARK DATASETS

For benchmark datasets, we compare the tube loss function with QD loss function-based NN models using the same experimental setup followed in Pearce et al. (2018) except the choice of optimizer.

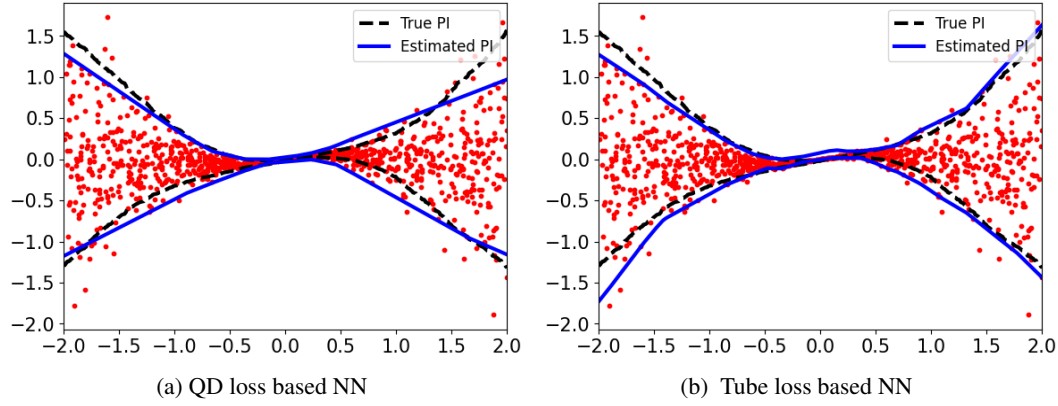

(a) QD loss based NN        (b) Tube loss based NN

Figure 7

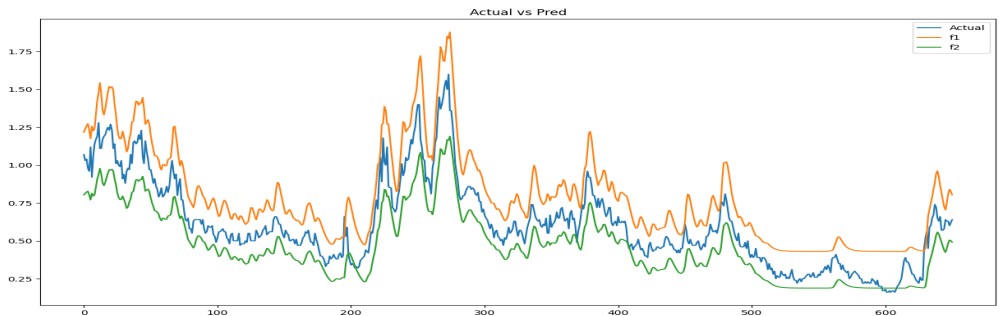

Figure 8: Probablistic forecast of LSTM with proposed tube loss function.

For the tube loss based NN model, we have used the optimizer 'RMSprop'. The NN models were asked to output $95\%$ PI and have used five NN models per ensemble.

The numerical results are reported in Table 3 for four benchmark datasets. For QD loss-based NN, we have used the numerical results reported in (Pearce et al. (2018)). The tube loss obtains a more accurate PI than the existing QD loss in the NN model.

### 4.3    TUBE LOSS IN DEEP LEARNING

We collect the hourly recordings of time-series significant ocean wave height from the https://www.ndbc.noaa.gov/ corresponding to the buoy station 42001 for 21 April 2021 - 25 July 2021. Our target is to obtain the hourly probabilistic forecast with $0.95$ confidence. For this, we construct the an LSTM model with three hidden layers with 128, 64 and 32 hidden neurons with drop-out layers. The window size is 100. For training $70\%$ of the data points are used and rest of them are used for testing.

We use the proposed tube loss in LSTM with two output neurons in final layer for obtaining the PI with confidence 0.95. The tube loss based LSTM model results in a PICP equal to 0.977 and MPIW equal to 0.307 with $\delta = 0$. But after recalibration with $\delta = 0.06$, we could obtain PICP equal to 0.947 with MPIW 0.294. Figure 8 shows the forecast of the LSTM model with tube loss function.

## 5    REPRODUCIBILITY STATEMENT

The numerical results and figures obtained with our proposed methods can be regenerated with the code and data provided in supplementary files.

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

## A  SUPPLEMENTARY MATERIAL
## GRADIENT DESCENT METHOD FOR SVPI MODEL

The Support Vector Prediction Interval (SVPI) model estimate functions

$$\mu_1(x) := \sum_{i=1}^{l} k(x_i, x)\alpha_i + b_1 \ \text{ and } \ \mu_2(x) := \sum_{i=1}^{l} k(x_i, x)\beta_i + b_2. \tag{14}$$

where $k(x, y)$ is positive definite kernel. For the sake of simplicity, we rewrite $\mu_1(x)$ and $\mu_2(x)$ in vector form

$$\mu_1(x) := K(A^T, x)\alpha + b_1 \ \text{ and } \ \mu_2(x) := K(A^T, x)\beta + b_2. \tag{15}$$

where $A$ is the $l \times n$ data matrix containing $l$ training points in $\mathbb{R}^n$, $K(A^T, x) = [k(x_1, x), k(x_2, x), .., k(x_l, x)]$, $\alpha = \begin{bmatrix} \alpha_1 \\ \alpha_2 \\ .. \\ \alpha_l \end{bmatrix}$ and $\beta \begin{bmatrix} \beta_1 \\ \beta_2 \\ .. \\ \beta_l \end{bmatrix}$. The SVPI considers the problem

$$\min_{(\alpha, \beta, b_1, b_2)} J_2(\alpha, \beta, b_1, b_2) =$$

$$\frac{\lambda}{2}(\alpha^T \alpha + \beta^T \beta) + \sum_{i=1}^{l} L_t\big(y_i, \big(K(A^T, x_i)\alpha + b_1\big), \big(K(A^T, x_i)\beta + b_2\big)\big) + \delta \sum_{i=1}^{l} \big|(K(A^T, x_i)(\alpha - \beta) + (b_1 - b_2)\big| \tag{16}$$

For a given point $(x_i, y_i)$, let us compute the gradient of $L_t\big(y_i, \big(K(A^T, x_i)\alpha + b_1\big), \big(K(A^T, x_i)\beta + b_2\big)\big)$ first. For this, we compute

$$\frac{\partial L_t\big(y_i, \big(K(A^T, x_i)\alpha+b_1\big), \big(K(A^T, x_i)\beta+b_2\big)\big)}{\partial \beta} =$$

$$\begin{cases} (1-t)K(A, x_i) & \text{,if } (K(A^T, x_i)\beta + b_2) < y_i < (K(A^T, x_i)\alpha + b_1) \text{ and } y_i > (K(A^T, x_i)(r\alpha + (1-r)\beta) + (rb_1 + (1-r)b_2). \\ 0 & \text{,if } (K(A^T, x_i)\beta + b_2) < y_i < (K(A^T, x_i)\alpha + b_1) \text{ and } y_i < (K(A^T, x_i)(r\alpha + (1-r)\beta) + (rb_1 + (1-r)b_2). \\ 0 & \text{,if } K(A^T, x_i)\beta + b_2 > y. \\ -tK(A^T, x_i) & \text{,if } K(A^T, x_i)\alpha + b_1 < y. \end{cases}$$

$$\frac{\partial L_t\big(y_i, \big(K(A^T, x_i)\alpha+b_1\big), \big(K(A^T, x_i)\beta+b_2\big)\big)}{\partial b_2} =$$

$$\begin{cases} (1-t) & \text{,if } (K(A^T, x_i)\beta + b_2) < y_i < (K(A^T, x_i)\alpha + b_1) \text{ and } y_i > (K(A^T, x_i)(r\alpha + (1-r)\beta) + (rb_1 + (1-r)b_2). \\ 0 & \text{,if } (K(A^T, x_i)\beta + b_2) < y_i < (K(A^T, x_i)\alpha + b_1) \text{ and } y_i < (K(A^T, x_i)(r\alpha + (1-r)\beta) + (rb_1 + (1-r)b_2). \\ 0 & \text{,if } K(A^T, x_i)\beta + b_2 > y. \\ -t & \text{,if } K(A^T, x_i)\alpha + b_1 < y. \end{cases}$$

$$\frac{\partial L_t\big(y_i, \big(K(A^T, x_i)\alpha+b_1\big), \big(K(A^T, x_i)\beta+b_2\big)\big)}{\partial \alpha} =$$

$$\begin{cases} 0 & \text{,if } (K(A^T, x_i)\beta + b_2) < y_i < (K(A^T, x_i)\alpha + b_1) \text{ and } y_i \geq (K(A^T, x_i)(r\alpha + (1-r)\beta) + (rb_1 + (1-r)b_2). \\ -(1-t)K(A, x_i) & \text{,if } (K(A^T, x_i)\beta + b_2) < y_i < (K(A^T, x_i)\alpha + b_1) \text{ and } y_i < (K(A^T, x_i)(r\alpha + (1-r)\beta) + (rb_1 + (1-r)b_2). \\ tK(A, x_i) & \text{,if } K(A^T, x_i)\beta + b_2 > y. \\ 0 & \text{,if } K(A^T, x_i)\alpha + b_1 < y. \end{cases}$$

$$\frac{\partial L_t\big(y_i, \big(K(A^T, x_i)\alpha+b_1\big), \big(K(A^T, x_i)\beta+b_2\big)\big)}{\partial b_1} =$$

$$\begin{cases} 0 & \text{,if } (K(A^T, x_i)\beta + b_2) < y_i < (K(A^T, x_i)\alpha + b_1) \text{ and } y_i \geq (K(A^T, x_i)(r\alpha + (1-r)\beta) + (rb_1 + (1-r)b_2). \\ -(1-t) & \text{,if } (K(A^T, x_i)\beta + b_2) < y_i < (K(A^T, x_i)\alpha + b_1) \text{ and } y_i < (K(A^T, x_i)(r\alpha + (1-r)\beta) + (rb_1 + (1-r)b_2). \\ t & \text{,if } K(A^T, x_i)\beta + b_2 > y. \\ 0 & \text{,if } K(A^T, x_i)\alpha + b_1 < y. \end{cases}$$

**Note:-** For point $(x_k, y_k)$ such that $y_k = K(A^T, x_k)\alpha + b_1$, $K(A^T, x_k)\beta + b_2$ or $K(A^T, x_k)(r\alpha + (1-r)\beta) + (rb_1 + (1-r)b_2)$, the tube loss function is not smooth, hence unique gradient do not exist on these points. At these points, any sub-gradient can be considered.

Now, for given data point $(x_i, y_i)$, we consider the width of PI tube $\delta\big|(K(A^T, x_i, y_i)(\alpha - \beta) + (b_1 - b_2)\big|$ and denote it as $J(\alpha, \beta, b_1, b_2, x_i, y_i)$ and compute

$$\frac{\partial J(\alpha, \beta, b_1, b_2, x_i)}{\partial \alpha} = sign((K(A^T, x_i)(\alpha - \beta) + (b_1 - b_2))K(A^T, x_i) \quad \text{if } (K(A^T, x_i)(\alpha - \beta) + (b_1 - b_2) \neq 0$$

$$\frac{\partial J(\alpha, \beta, b_1, b_2, x_i)}{\partial b_1} = sign((K(A^T, x_i)(\alpha - \beta) + (b_1 - b_2)) \quad \text{if } (K(A^T, x_i)(\alpha - \beta) + (b_1 - b_2) \neq 0$$

$$\frac{\partial J(\alpha, \beta, b_1, b_2, x_i)}{\partial \beta} = -sign((K(A^T, x_i)(\alpha - \beta) + (b_1 - b_2))K(A^T, x_i) \quad \text{if } (K(A^T, x_i)(\alpha - \beta) + (b_1 - b_2) \neq 0$$

$$\frac{\partial J(\alpha, \beta, b_1, b_2, x_i)}{\partial b_2} = -sign((K(A^T, x_i)(\alpha - \beta) + (b_1 - b_2)) \quad \text{if } (K(A^T, x_i)(\alpha - \beta) + (b_1 - b_2) \neq 0$$

For data point $(x_i, y_i)$ satisfying $(K(A^T, x_i)(\alpha - \beta) + (b_1 - b_2) = 0$ , unique gradient do not exist and any sub gradient can be considered.

Now, we state gradient descent algorithm for SVPI problem.
**Algorithm 2:-**

  Input:- Training Set $T = \{(x_i, y_i) : x_i \in \mathbb{R}^n, y_i \in \mathbb{R}, i = 1, 2, ...l\}$, confidence $t \in (0, 1)$ $r, \delta, \eta$ and $tol$.
  Initialize:- $\alpha^0, \beta^0 \in \mathbb{R}^l$ and $b_1^0, b_2^0 \in \mathbb{R}$.

  Repeat

$$\beta^{(k+1)} = \beta^{(k)} - \eta_k\Big(\lambda\beta^{(k)} + \sum_{i=1}^{l}\frac{\partial L_t\Big(y_i,\big(K(A^T,x_i)\alpha+b_1\big),\big(K(A^T,x_i)\beta+b_2\big)\Big)}{\partial\beta^{(k)}} +$$

$$\delta\sum_{i=1}^{l}\frac{\partial J(\alpha,\beta,b_1,b_2,x_i,y_i)}{\partial\beta^{(k)}}\Big)$$

$$b_2^{(k+1)} = b_2^{(k)} - \eta_k\Big(\sum_{i=1}^{l}\frac{\partial L_t\Big(y_i,\big(K(A^T,x_i)\alpha+b_1\big),\big(K(A^T,x_i)\beta+b_2\big)\Big)}{\partial b_2^{(k)}} + \delta\sum_{i=1}^{l}\frac{\partial J(\alpha,\beta,b_1,b_2,x_i,y_i)}{\partial b_2^{(k)}}\Big)$$

$$\alpha^{(k+1)} = \alpha^{(k)} - \eta_k\Big(\lambda\alpha^{(k)} + \sum_{i=1}^{l}\frac{\partial L_t\Big(y_i,\big(K(A^T,x_i)\alpha+b_1\big),\big(K(A^T,x_i)\beta+b_2\big)\Big)}{\partial\alpha^{(k)}} +$$

$$\delta\sum_{i=1}^{l}\frac{\partial J(\alpha,\beta,b_1,b_2,x_i,y_i)}{\partial\alpha^{(k)}}\Big)$$

$$b_1^{(k+1)} = b_1^{(k)} - \eta_k\Big(\sum_{i=1}^{l}\frac{\partial L_t\Big(y_i,\big(K(A^T,x_i)\alpha+b_1\big),\big(K(A^T,x_i)\beta+b_2\big)\Big)}{\partial b_1^{(k)}} + \delta\sum_{i=1}^{l}\frac{\partial J(\alpha,\beta,b_1,b_2,x_i,y_i)}{\partial b_1^{k}}\Big)$$

Until $\left\|\begin{bmatrix}\alpha^{k+1}-\alpha^k\\b_1^{k+1}-b_1^k\\\beta^{k+1}-\beta^k\\b_2^{k+1}-b_2^k\end{bmatrix}\right\| \geq tol.$

