# OpenReview forum: "Tube Loss: A Novel Approach for High Quality Prediction Interval Estimation"
_ICLR.cc/2024/Conference — ICLR 2024 Conference Withdrawn Submission_

### Official Review · Reviewer_PHGF · 2023-10-31

**Soundness:** 3 good
**Presentation:** 2 fair
**Contribution:** 3 good
**Rating:** 3
**Confidence:** 3

**Summary:**

This work propose a continuous loss, named tube loss, for prediction interval estimation. The loss is a Piecewise linear function with respect to the predictions. It optimizes the upper and lower functions of the prediction interval to find the proper confidence space and minimize the width of the space simultaneously. The effectiveness of the proposed loss is verified both theoretically and experimentally.

**Strengths:**

The optimization of the prediction interval and the minimization of the width are combined into the same objective, which is much simpler than existing methods. It introduces a tuning parameter r to separate the prediction interval into two parts, r can be adjusted to enable the PI tube to capturing the denser regions of the prediction, and thus reducing the width.
the method is claimed to be able to cover the densest regions of response values, thus allowing the sharpening of the width of PI, especially when the distribution of the response is skewed.

**Weaknesses:**

The optimization is a two-stage process. The setting of r and lambda is not automatically, need human experience.
The manuscript needs more serious revision, some equations contain undefined variables, some figures have no captions, and some figures are not explained/referred in the main material.
the effectiveness of the proposed method on situations where the distribution of the response is skewed is not clearly shown in the experimental part.

**Questions:**

1.	The equation on page one need to be rectified, E(y/x)?.
2.	Equation(8), there is no i in the second summation, and what is l for?
3.	How many iterations are used in the optimization? How to find a proper tuning parameter r for a new dataset?
4.	Figure 1 is not referred in the main material. Figure 6, 7 have no captions.
6.	Table 3, the proposed method is not always better than QD, e.g. on data Energy, the performance of QD is better than Tube loss, and the MPIW measure of QD is better than Tube loss, the authors should provide a detailed discussion on these results.
7.	Section 4.2, what kind of neural network is used in the experiments? Please provide a description on the architecture.
8.	The tile for section 4.3 needs revision

---

### Official Review · Reviewer_9f1c · 2023-11-01

**Soundness:** 2 fair
**Presentation:** 1 poor
**Contribution:** 1 poor
**Rating:** 3
**Confidence:** 2

**Summary:**

The authors propose a new loss function named tube loss for prediction interval estimation. The authors also demonstrate the effectiveness of tube loss by numerical experiments.

**Strengths:**

The prediction interval estimation problem addressed in this paper might be a relatively significant issue.

**Weaknesses:**

1. The writing style of this article does not meet the standards expected for academic publications.
2. This paper fails to elucidate certain crucial symbols and concepts, such as the function $E$ and $\mu$ mentioned in the first paragraph. 3. The paper lacks a section on related work, making it challenging for me to discern the challenges addressed and the contributions made by this manuscript.
4. The experimental section is rather cursory and lacks detailed descriptions of many aspects.

**Questions:**

What are the primary challenges addressed by the article? And what are its main contributions?

---

### Official Review · Reviewer_tfsf · 2023-11-03

**Soundness:** 1 poor
**Presentation:** 1 poor
**Contribution:** 1 poor
**Rating:** 1
**Confidence:** 5

**Summary:**

The paper proposes a loss function that allows estimating prediction intervals. The authors propose parametrizing the lower and upper “quantiles” of the prediction interval via kernel machines, and simultaneously learn them while reducing the expected interval width.

**Strengths:**

- The paper tackles an important problem: estimating reliable prediction intervals.
- The proposed loss function seems interesting.

**Weaknesses:**

- The paper is written quite poorly, badly organized etc.
- The proposed loss function is not motivated enough.
- Baselines with quantile-regression–based interval estimation are missing.
- Experiments on real data are limiting and do not indicate that the proposed method is better.

**Questions:**

- _The loss function._ The proposed loss function, while seems interesting, looks arbitrary – i.e., the authors do not motivate it well. I appreciate the authors doing extensive experiments studying the effect of the parameter r. But I would encourage the authors to motivate the loss function better before introducing it. This makes the reader appreciate it better.
- _Quantile regression._ The de-facto approach for producing confidence intervals is quantile regression. However, it seems that the authors neither compare to nor discuss even a single work related to quantile regression. This makes me wonder if the authors are even familiar with this large body of work. I would encourage the authors to add a discussion and a comparison to quantile regression. This helps the reader understand the context of this work in the literature.
- _Real data experiments._ While I appreciate the synthetic data experiments and the exploration of the effect of the parameter r, the experiments on real data are limited. Moreover, the experiments on real data (Table 3) suggests that the proposed loss function produces consistently larger width QD loss (a prior work). The authors mention that this is because the QD loss does not reach desired level of coverage: but I believe this is inaccurate characterization, consider for e.g. $\texttt{Concrete}$ and $\texttt{Energy}$ datasets, they produce about the same coverage, but significantly lower width. This indicates that QD loss is better than the proposed loss function. On the other two datasets, since QD loss does not achieve the same coverage, it is _not comparable_ to the proposed loss function, but this does *not* indicate that the proposed loss function performs better.
- _Writing, organization etc._ The paper is written poorly overall. The proofs and Page 6 do not contribute much to the paper. I would rather see this space utilized for more real-data experiments.

Overall, due to the above limitations, I believe the paper is not up to the standard of the conference. I encourage the authors to incorporate the aforementioned suggestions, it might make the paper better.

---

### Official Review · Reviewer_RfXE · 2023-11-03

**Soundness:** 2 fair
**Presentation:** 2 fair
**Contribution:** 2 fair
**Rating:** 3
**Confidence:** 3

**Summary:**

The authors propose a new loss function that can simultaneously learn upper and lower confidence intervals. The authors show that in the limit of samples $ \to \infty$, the proposed loss function will yield prediction intervals for $t$ fraction of the values. Experiments are run on kernel machines, LSTMs, on synthetic data as well as real world time series data ( ocean wave height, as well as benchmark datasets from Pearce et al 2018).

**Strengths:**

- The suggested loss function simultaneously learns both boundaries of the prediction interval, which allows for minimizing the mean average width.

- The loss is continuous and differentiable almost everywhere, which allows for using modern optimization algorithms.

- Experiments show improvements over the QD loss proposed in Pearce et al 2018.

**Weaknesses:**

- It's not clear what the main strength of this paper is. The propositions themselves are not very technically novel, the loss function is a modification of the Huber loss, with four pieces instead of two, and the experiments are either on synthetic data or compared to a paper from 2018.

- The baseline comparisons are weak -- the authors only consider a paper from 2018 by Pearce et al, and the claim is that in table 3, the intervals are more accurate in the proposed approach. (it's also not clear how to interpret Table 3 -- what's the verification that TUBE loss is better than QD loss?)

- The experiments are mostly on synthetic data, except for the comparison to Pearce et al, and experiments without baselines on the Ocean wave height dataset ( section 4.3).

**Questions:**

See the weaknesses section

---

### Meta-Review · Area_Chair_6gNn · 2023-12-18

**Metareview:**

All reviewers unanimously point flaws in the paper, including the ad hoc formulation of the loss without any justification, potential mistakes or typoes in the draft, the weakness of the baselines for comparison, limitation in the experiments, and more general organisation comments (no related work section, general presentation).

The authors have not submitted any rebuttal. It is recommended they use the reviews to improve on the work presented.

**Justification For Why Not Higher Score:**

Very weak paper.

**Justification For Why Not Lower Score:**

N/A

---

### Decision · Program_Chairs · 2024-01-16

Reject